# Non-Idealities in Lab-Scale Kinetic Testing: A Theoretical Study of a Modular Temkin Reactor

Gregor D. Wehinger [1,*] , Bjarne Kreitz [2] and C. Franklin Goldsmith [2]

1 Institute of Chemical and Electrochemical Process Engineering, Clausthal University of Technology, 38678 Clausthal-Zellerfeld, Germany

2 School of Engineering, Brown University, Providence, RI 02912, USA; bjarne_kreitz@brown.edu (B.K.); franklin_goldsmith@brown.edu (C.F.G.)

* Correspondence: wehinger@icvt.tu-clausthal.de; Tel.: +49-5323-722183

**Abstract:** The Temkin reactor can be applied for industrial relevant catalyst testing with unmodified catalyst particles. It was assumed in the literature that this reactor behaves as a cascade of continuously stirred tank reactors (CSTR). However, this assumption was based only on outlet gas composition or inert residence time distribution measurements. The present work theoretically investigates the catalytic $CO_2$ methanation as a test case on different catalyst geometries, a sphere, and a ring, inside a single Temkin reaction chamber under isothermal conditions. Axial gas-phase species profiles from detailed computational fluid dynamics (CFD) are compared with a CSTR and 1D plug-flow reactor (PFR) model using a sophisticated microkinetic model. In addition, a 1D chemical reactor network (CRN) model was developed, and model parameters were adjusted based on the CFD simulations. Whereas the ideal reactor models overpredict the axial product concentrations, the CRN model results agree well with the CFD simulations, especially under low to medium flow rates. This study shows that complex flow patterns greatly influence species fields inside the Temkin reactor. Although residence time measurements suggest CSTR-like behavior, the reactive flow cannot be described by either a CSTR or PFR model but with the developed CRN model.

**Keywords:** catalyst testing; temkin reactor; CFD; $CO_2$ methanation; reactor modeling; chemical reactor network

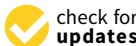

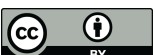

## 1. Introduction

The determination of intrinsic reaction kinetics is crucial for any catalyst development program. However, in gas–solid reaction systems, internal and external heat and mass transport limitations can occur and, therefore, affect the observed reaction rates. The absence of these phenomena can be achieved by guaranteeing the following conditions [1]: effective contacting between the fluid reactants and the catalyst, no internal and external heat and mass transport limitations, as well as ideal flow (either plug-flow reactor, PFR, or continuously stirred tank reactor, CSTR), and isothermal conditions. The absence of transport limitations is achieved by a small catalyst particle size and high flow rates, while the type of reactor affects the flow and thermal conditions. Typical lab-scale reactors mimicking CSTR conditions are Berty- and Carberry-type reactors [2,3]. High conversion conditions and original catalyst particle dimensions are desirable for industrial purposes. This becomes extremely challenging in the case of egg-shell or egg-yolk catalyst particles, which cannot be crushed to smaller particles, and testing of the actual catalyst geometries is necessary [4–8]. The Temkin reactor concept tries to accomplish these goals. In the original Temkin reactor, large catalyst pellets were alternated with inert particles in order to achieve isothermal conditions and defined plug-flow-like behavior [9]. Another design is the "single pellet string reactor", where the tube-to-pellet diameter ratio is between 1.1 and 1.4. This concept was applied for spherical particles [10,11], cylinders [12], extrudates [13], or multi-hole cylinders [14] to name only a few examples. Several theoretical studies based

on computational fluid dynamics (CFD) revealed that the single pellet string reactor design does not behave as a perfect PFR in terms of residence time distribution, and isothermal conditions might not be maintained [15,16].

Nonetheless, further development of the original Temkin reactor was carried out in the 2010s with the idea of realizing a cascade of CSTRs that include one single catalyst pellet. Vinyl acetate monomer catalysts were tested in a modified Temkin version from ILS [17], while at TU Darmstadt the selective hydrogenation of acetylene was studied in another Temkin reactor design [18,19]. This so-called "advanced Temkin reactor" design is composed of modules of two half-shells of a stainless-steel cylinder with a large number of cavities in which the pellets are inserted. The same authors also presented a CFD study, where almost isothermal conditions ($\Delta T < 2$ K) in egg-shell catalysts were confirmed for the selective hydrogenation of acetylene [20].

The local flow field can be studied in great detail with CFD simulations, while considering the microkinetics of the surface reactions. Local interactions between transport phenomena, i.e., heat and mass transport, and reaction rates occur and are dependent on the reactor geometry, operating conditions, catalyst material, and reaction kinetics [21]. This was demonstrated for catalytic honeycomb reactors [22], foam monolith reactors [23], or packed bed reactors [24,25]. Especially, the resolution of fine details of such complex structures requires a large number of computational cells in the finite-volume method framework or even local geometric modifications [26,27]. Since detailed CFD studies of catalytic reactors are time consuming, it is desirable for reaction engineers to describe them with simpler models, e.g., PFR or CSTR models with or without non-idealities. For catalytic testing reactors, it must be noted that, depending on the underlying reactor model, different kinetic parameters can be obtained for the same experimental results [28]. Consequently, we studied in previous work the flow behavior of a modified Temkin reactor, where one single catalyst pellet is placed in a small chamber by combining CFD simulations with residence time distribution (RTD) measurements [16]. Spherical and ring-shaped catalyst particles were tested, and the resulting RTDs were compared with a tanks-in-series model. That study showed that the modular Temkin reactor from ILS behaves as an almost ideal CSTR concerning the RTD. Depending on the volumetric flow rate and the particle shape, the RTD can be satisfactorily described with three or fewer tanks-in-series, corresponding to a Bodenstein number ≤3. Nonetheless, the local flow fields inside the reactor chamber are complex, with recirculation, channeling, and stagnant zones. Finally, mean-age-of-air CFD simulations highlighted that those stagnant zones close to the inlet of the single chamber significantly affect the residence time [16].

In this work, we aim to extend our previous study [16] by including the hydrogenation of carbon dioxide, also known as the Sabatier reaction, in the CFD model. Based on detailed isothermal CFD simulations, we discuss the hypothesis that the gas-phase concentration profiles of the modified Temkin reactor can be described with simplified reactor models (CSTR, PFR, and a combination of those) including the chemical reaction on the outer surface of a ring and a spherical catalyst geometry. Therefore, we analyze the flow conditions, as well as gas phase and surface species concentration distributions, while neglecting internal heat and mass transport limitations. The findings of this study can help reaction engineers apply appropriate but simple and fast reactor models for the modular Temkin reactor to finally develop intrinsic reaction kinetics.

## 2. Materials and Methods

### 2.1. Computational Fluid Dynamics

#### 2.1.1. Governing Equations

The governing equations comprise conservation of total mass, conservation of momentum, conservation of chemical species mass, and conservation of energy in terms of specific enthalpy. Specific details on chemically reacting flow in the CFD framework with an emphasis on surface reactions can be found elsewhere [24,29]. The turbulent Reynolds Averaged Navier Stokes (RANS) equations read:

Conservation of mass:

$$\nabla \cdot (\rho \bar{\mathbf{v}}) = 0 \tag{1}$$

where $\rho$ is the mass density and $\bar{\mathbf{v}}$ is the mean velocity vector.

Conservation of momentum:

$$\nabla \cdot (\rho \bar{\mathbf{v}} \bar{\mathbf{v}}) = -\nabla \cdot \bar{p} \mathbf{I} + \nabla \cdot (\mathbf{T} + \mathbf{T}_{\text{RANS}}) \tag{2}$$

where $\rho$ is the fluid density, $\bar{\mathbf{v}}$ and $\bar{p}$ are the mean velocity and pressure, respectively, and $\mathbf{I}$ is the identity tensor, $\mathbf{T}$ is the viscous stress tensor, defined for Newtonian fluids as:

$$\mathbf{T} = -2/3\mu(\nabla \cdot \bar{\mathbf{v}})\mathbf{I} + 2\mu\mathbf{D} \tag{3}$$

where $\mu$ is the dynamics viscosity of the fluid and $\mathbf{D}$ is the deformation, which is defined as:

$$\mathbf{D} = \frac{1}{2}\left[\nabla\bar{\mathbf{v}} + (\nabla\bar{\mathbf{v}})^T\right] \tag{4}$$

The RANS stress tensor $\mathbf{T}_{\text{RANS}}$ reads:

$$\mathbf{T}_{\text{RANS}} = -\begin{pmatrix} \rho\overline{v'_x v'_x} & \rho\overline{v'_x v'_y} & \rho\overline{v'_x v'_z} \\ \rho\overline{v'_y v'_x} & \rho\overline{v'_y v'_y} & \rho\overline{v'_y v'_z} \\ \rho\overline{v'_z v'_x} & \rho\overline{v'_z v'_y} & \rho\overline{v'_z v'_z} \end{pmatrix} + \frac{2}{3}\rho k\mathbf{I} \tag{5}$$

where $k$ is the turbulent kinetic energy. This tensor is modeled as a function of mean flow quantities, i.e., the Boussinesq approximation:

$$\mathbf{T}_{\text{RANS}} = -2/3\mu_{\text{t}}(\nabla \cdot \bar{\mathbf{v}})\mathbf{I} + 2\mu_{\text{t}}\mathbf{D} \tag{6}$$

with $\mu_{\text{t}}$ as the turbulent eddy viscosity of the fluid. In this study, the realizable $k$-$\varepsilon$ RANS turbulence model is used with an All $y^+$ wall-treatment. This specific RANS turbulence model expresses the critical coefficient $C_\mu$ as a function of mean flow and turbulence properties, rather than being constant.

Conservation of species mass $i$ without homogeneous gas phase reactions:

$$\nabla \cdot (\rho \bar{\mathbf{v}} \bar{Y}_i) + \nabla \cdot \mathbf{j}_i = 0 \quad \text{for} \quad i = 1, \dots, N_{\text{g}} \tag{7}$$

with the mean mass fraction $\bar{Y}_i = m_i/m$ of species $i$ and total mass $m$, and $N_{\text{g}}$ is the number of gas phase species. The diffusive mass flux components $\mathbf{j}_i$ are mixture-average defined:

$$\mathbf{j}_i = -\rho \frac{\bar{Y}_i}{\bar{X}_i} D_i^{\text{M}} \nabla \bar{X}_i \tag{8}$$

where $D_i^{\text{M}}$ is the effective diffusion coefficient between species $i$ and the mixture M:

$$D_i^{\text{M}} = \frac{1 - \bar{Y}_i}{\sum_{j \neq i}^{N_{\text{G}}} \bar{X}_j / D_{ij}} \qquad \text{for } i = 1, \dots, N_{\text{G}} \tag{9}$$

The binary diffusion coefficients $D_{ij}$ are governed from kinetic gas theory via Chapman–Enskog [30]. The mean molar fraction $\bar{X}_i$ can be written as:

$$\bar{X}_i = \frac{1}{\sum_{j=1}^{N_{\text{g}}} \frac{\bar{Y}_j}{M_j}} \frac{\bar{Y}_i}{M_i} \tag{10}$$

with $M_i$ as the molecular weight of species $i$.

Conservation of energy in terms of specific enthalpy $h$ in the gas phase without homogeneous gas-phase reactions:

$$\nabla \cdot \left( \rho \bar{\mathbf{v}} \bar{h} \right) + \nabla \cdot \mathbf{j}_q = \nabla \cdot (\bar{\mathbf{v}} \bar{p}) - \nabla \cdot (\mathbf{T} + \mathbf{T}_{\text{RANS}}) \bar{\mathbf{v}} \tag{11}$$

with the diffusive heat transport $\mathbf{j}_q$ given by:

$$\mathbf{j}_q = -k \nabla \bar{T} + \sum_{i=1}^{N_g} \bar{h}_i \mathbf{j}_i \tag{12}$$

with the thermal conductivity of the mixture $k$ and the mean mixture specific enthalpy $\bar{h}$ as a function of temperature $\bar{h} = \bar{h}(\bar{T})$:

$$\bar{h} = \sum_{i=1}^{N_g} \bar{Y}_i \bar{h}_i(\bar{T}) \tag{13}$$

Here, the *realizable k-ε* RANS turbulence model [31] with an *all $y^+$ wall-treatment* is used for high flow rates. This model defines the critical coefficient $C_\mu$ as a function of mean flow and turbulence properties. Ideal gas was assumed connecting pressure, temperature, and density to close the governing equations. For more details about species transport equations and properties, please see, e.g., ref. [32].

### 2.1.2. Modeling Surface Reactions

The mean-field approximation is applied in this study to describe the catalytic chemistry on the surface with microkinetic models [29,33]. Therefore, the model assumes uniformly distributed catalytic sites and adsorbates across a computational cell face. Under steady-state conditions, gas-phase molecules of species $i$, which are consumed/produced at the reactive surface by adsorption/desorption, have to diffuse from/to the catalyst surface [33]:

$$\mathbf{j}_i = R_i^{\text{het}} \tag{14}$$

with the heterogeneous reaction term $R_i^{\text{het}}$ as:

$$R_i^{\text{het}} = F_{\text{cat/geo}} M_i s_i \tag{15}$$

where $s_i$ is the molar net production rate of gas-phase species $i$ and $F_{\text{cat/geo}}$ is the ratio between the catalytic active area $A_{\text{cat}}$ and the geometric area $A_{\text{geo}}$:

$$F_{\text{cat/geo}} = A_{\text{cat}} / A_{\text{geo}} \tag{16}$$

$$s_i = \sum_{k=1}^{K_s} v_{ik} k_{f_k} \prod_{j=1}^{N_g + N_{\text{ads}}} c_j^{v'_{jk}} \tag{17}$$

with $K_s$ as the number of surface reactions, and $c_j$ as the species concentrations, either for the adsorbed species $N_{\text{ads}}$ or for the gas-phase species $N_g$, respectively. In addition, the surface coverage $\Theta$ considers the surface site density $\Gamma$ (mol m$^{-2}$) giving the maximum number of species adsorbing on a unit surface area. The coordinate number $\varsigma_i$ represents the number of surface sites occupied by the species $i$, where $c_i$ is the concentration of adsorbed species in units mol m$^{-2}$:

$$\Theta_i = c_i \varsigma_i \Gamma^{-1} \tag{18}$$

The forward reaction rate $k_{f_k}$ is expressed with a modified Arrhenius function taking a temperature dependence into account ($\beta_k$):

$$k_{f_k} = A_k T^{\beta_k} \exp\left(\frac{-E_{a_k}}{RT}\right) \tag{19}$$

For adsorption reactions, the rate coefficient is expressed with sticking coefficients $\sigma$:

$$k_{f_k}^{ads} = \frac{\sigma_i^0}{\Gamma^\tau} \sqrt{\frac{RT}{2\pi M_i}} \tag{20}$$

with $\sigma_i^0$ as the initial (uncovered surface) sticking coefficient:

$$\sigma_i^0 = a T^b \exp\left(\frac{-c}{RT}\right) \tag{21}$$

where $a$, $b$, and $c$ are constants specific for the reaction $k$. $\tau = \sum_{j=1}^{N_s} \nu'_{jk}$ is the sum of all the surface reactant's stoichiometric coefficients [32,33]. Thermodynamic consistency of the mechanism is maintained by defining the reverse reaction rate $k_r$ via the forward reaction rate $k_f$ and the equilibrium constant $K_c$:

$$k_{r_k} = \frac{k_{f_k}}{K_{c_k}} \tag{22}$$

The equilibrium constant in concentration units are determined from the thermodynamic properties in pressure units:

$$K_{c_k} = K_{p_k} \left(\frac{p_{atm}}{RT}\right)^{\sum_{j=1}^{N} \nu_{jk}} \tag{23}$$

whereas the equilibrium constant $K_{p_k}$ is obtained with the relationship:

$$K_{p_k} = \exp\left(\frac{\Delta S_k^0}{R} - \frac{\Delta H_k^0}{RT}\right) \tag{24}$$

The $\Delta$ represents the change that occurs in passing completely from reactants to products. Heat capacity $C_p^0$, entropy $S^0$, and enthalpy $H^0$ are expressed as polynomials in the NASA 7-coefficient polynomial parameterization, where seven coefficients are needed for each of two temperature ranges:

$$\frac{c_p^0(T)}{R} = a_0 + a_1 T + a_2 T^2 + a_3 T^3 + a_4 T^4 \tag{25}$$

$$\frac{H^0(T)}{RT} = a_0 + \frac{a_1}{2} T + \frac{a_2}{3} T^2 + \frac{a_3}{4} T^3 + \frac{a_4}{5} T^4 + \frac{a_5}{T} \tag{26}$$

$$\frac{S^0(T)}{R} = a_0 \ln(T) + a_1 T + \frac{a_2}{2} T^2 + \frac{a_3}{3} T^3 + \frac{a_4}{4} T^4 + a_6 \tag{27}$$

Other thermodynamic properties are easily given in terms of $c_p^0$, $S^0$, and $H^0$ [32].

### 2.1.3. CFD Setup

Only the flow-through volume of the Temkin reactor is taken into account for the CFD simulations. The catalyst particles, i.e., either 4 mm sphere or 3.6 mm oD ring, were placed close to the outlet of the chamber. Figure 1 shows the CAD drawing, the extracted gas-phase region, a detail of the computational mesh, as well as the spherical and ring particles. The computational mesh consists of polyhedral cells in the bulk region of the

domain and of prism cells close to the wall. The cell count is approx. 900,000. A mesh dependence study was already conducted in our previous study. Refer to ref. [16] for the details.

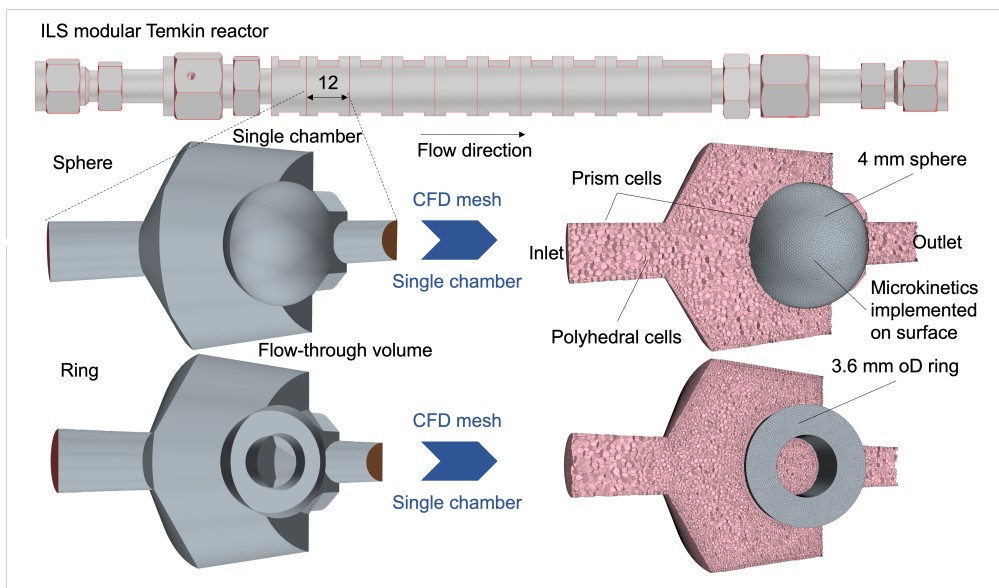

**Figure 1.** CFD setup.

A constant velocity was set as inlet boundary conditions in accordance to the norm volumetric flow rates (at $T = 273.15$ K): $\dot{V} = 50, 100, 200, 2000$ ml$_N$ min$^{-1}$. At the outlet, ambient pressure and the outflow boundary conditions $\partial \mathbf{v}/\partial n = 0$ were implemented. The reaction rates are described by the implemented microkinetic mechanism on the pellet surface. The segregated flow solver was applied to converge the set of steady-state equations. All CFD simulations were carried out with Simcenter STAR-CCM+ 2021.03 from Siemens [34]. In Simcenter STAR-CCM+, the reacting species transport equations are solved using the CVODE solver with the operator splitting algorithm to find an average reaction rate to remove stiffness. Different convergence criteria were set, i.e., pressure drop over the reactor chamber, outlet gas-phase species, and surface species. The solver requires about 50,000–100,000 iteration steps depending on the flow conditions. This was mainly due to the complex microkinetic model, since the flow field already converged at about 1000 iteration steps. In this study, no chemical acceleration method was used, although this would significantly speed up the calculations, as was shown recently elsewhere [35].

## 2.2. Simplification of Fluid Dynamics

A chemical engineering model consisting of ideal reactors is desirable for the modular Temkin reactor in order to replace the computationally expensive CFD model. The two common ideal reactors in chemical engineering are the CSTR and the PFR. With a smart combination of CSTRs, PFRs, and mixers, the flow field can be approximated with an arrangement of reactors, also known as chemical reactor networks (CRN), in which chemical kinetics can be locally included. In the past, CRNs were applied in combustion engineering of gas turbines [36], residence time distributions of flames [37], and emission predictions from pulverized coal flames [38] to name a few. Of particular interest is the development of CRNs based on detailed CFD simulations [39]. In one such study, the authors approximated the complex fluid dynamics of a gas turbine combustor with a CRN consisting of 22 chemical reactor elements, which were determined from the temperature and flame area density distribution and connected with split flows [40]. In another work, the gas turbine flame tube was approximated with up to 369 CSTRs, including a detailed chemical kinetic model, which showed an excellent agreement with the comprehensive CFD results [41]. An optimized procedure splits the flow field into homogeneous zones of CSTRs. In general,

with an increasing number of reactor elements, the CRN gains in accuracy in comparison with the CFD results, but the number of adjusted parameters increases, too. CRNs are typically found in the combustion community. The most prominent example for the application of these CRN models in the field of chemical reaction engineering are the compartment models often used to describe the residence time distribution of real reactors [42], e.g., in aerated reactors with multiple impellers [43] or stirred tank bioreactors [44,45]. Yet, for chemically reacting flow, similar approaches have been rarely applied. For example, mixing cell models were used to describe catalytic packed-bed reactors consisting of a discrete structure of CSTRs and PFRs [46]. Each structure has the dimensions of one diameter of a particle in a bed. However, no significant development has been carried out since the 1990s [47].

In this work, the fluid dynamics of one Temkin reactor chamber are simplified and modeled with a CRN according to Figure 2A,B. The flow enters the reaction chamber through a pipe from the left-hand side. While approaching the catalyst particle, some of the fluid recirculates in the upper-left section of the chamber. The size of the recirculation zone depends on the particle shape as well as on the volumetric flow rate, as was already shown in ref. [16]. The gas-phase species flow over the catalytic surface and react to the products according to the chemical kinetics. At the end of the reaction chamber, the fluid flows through a pipe and into the subsequent chamber (not shown here). The entrance (in blue) and exit (in black) are modeled as PFRs without chemical reactions, i.e., inert PFRs. The length $L$ of the PFR at the entrance is considered as a variable. The flow over the catalyst particle is modeled as a PFR with a reactive wall (red). Due to the complex reaction chamber geometry, the flow over the catalyst particle is not uniform. In other words, the full catalyst surface area is not available for the gas-phase reactants. Hence, the simplified model uses a modified available surface area $F'_{\text{cat/geo}}$. The recirculation zone is abstracted as a partial stream leaving the reactive PFR and feeding back into the inert PFR (blue) with a recycle ratio $R = \frac{\dot{m}_{\text{recycle}}}{\dot{m}_{\text{recycle}}}$ (yellow). As illustrated in Figure 2C, the recycle stream is mixed with the previous reactor stream and then enters the next mixer, which leads to an increase in the total mass flow rate after each mixer. This is repeated for all $N_{\text{inter reactors}}$. This model assumption evolved from the flow pattern inside the Temkin and aims to represent it accurately. The expansion of the eddy and consequently its effect is strong in the entrance of the chamber but decreases over the pellet due to the confinement of the flow channel. In total, a number of $N_{\text{reactors}} = 100$ was set, which leads to concentration profiles that are independent if $N_{\text{reactors}}$ are further increased. The impact of the back-mixing is strong, when the recycle ratio and the exit concentration is large. The three variables, $L$, $R$, $F'_{\text{cat/geo}}$, can be adjusted so that the axial gas-phase species profiles of the CFD model are predicted with the CRN model.

One-dimensional PFRs can be modeled essentially by a series of CSTRs in the axial direction (see Figure 2C). The steady-state solution of the PFR is obtained by performing a time integration to steady state for each CSTR in the chain. The state of the CSTR is then used as the inlet boundary condition for the next CSTR downstream. The recycle stream is fed back to each CSTR and mixed in a mixing chamber (also a CSTR) with the outlet concentration from the feed or the previous CSTR. This is justified since the eddy in Figure 2 extends almost over the entire pellet. Additional assumptions are no diffusive transport, constant pressure, and temperature. In this study, the open-source suite Cantera is used for the simplified reactor models [48]. The governing equations, i.e., mass conservation and species mass conservation, for a CSTR reads [32]:

$$\frac{dm}{dt} = \sum_{\text{in}} \dot{m}_{\text{in}} - \sum_{\text{out}} \dot{m}_{\text{out}} \tag{28}$$

$$m\frac{dY_k}{dt} = \sum_{\text{in}} \dot{m}_{\text{in}}(Y_{k,in} - Y_k) - \dot{m}_{Y,\text{wall}} \tag{29}$$

where the total (mass) production rate for the gas-phase species $k$ at the reactive wall is:

$$\dot{m}_{Y,\text{wall}} = W_k \sum_k A_{\text{wall}} \dot{s}_{k,\text{wall}} \qquad (30)$$

with $W_k$ as the molecular weight of species $k$, $A_{\text{wall}}$ as the wall area and $\dot{s}_{k,\text{wall}}$ as the molar rate of production for each homogeneous phase species $k$. For surface species $i$, the rate of change in surface coverage $\Theta_{i,\text{wall}}$ on each wall is integrated with time:

$$\frac{d\Theta_{i,\text{wall}}}{dt} = \frac{\dot{s}_{i,\text{wall}} \varsigma_i}{\Gamma} \qquad (31)$$

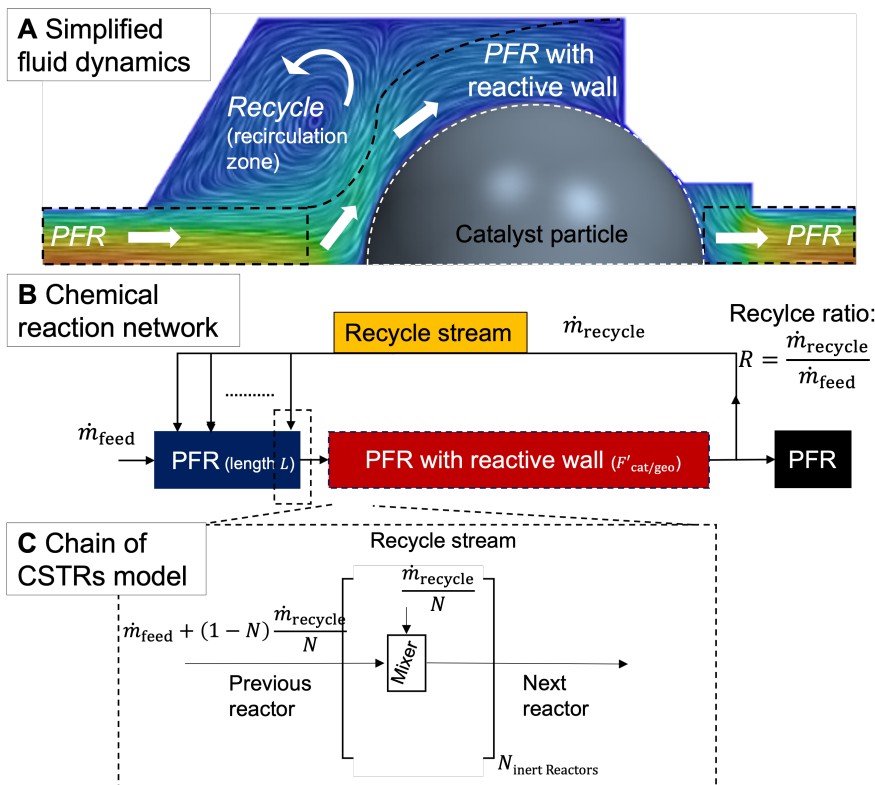

**Figure 2.** (**A**) Simplified fluid dynamics of one Temkin chamber and (**B**) the resulting CRN with a combination of PFRs and CSTRs. (**C**) Details on the chain of the CSTRs model. Back-mixing occurs only in the first PFR, where no reaction takes place.

The equations of the series of CSTRs are then solved by marching from the first to the last reactor, integrating each CSTR to a steady state. Relative and absolute tolerances on the simulations are $10^{-9}$ and $10^{-21}$, respectively. For further information on the open-source suite Cantera, see its website [48].

### 2.3. Carbon Dioxide Methanation

The $CO_2$ methanation is used as an example of fast reaction kinetics, which makes it challenging to model [49]. The microkinetic model was developed in a previous study from our groups. Briefly, the microkinetic model used in this study was constructed automatically by using the "Reaction Mechanism Generator" (RMG) [50–52], while considering the uncertainty in the DFT-derived databases of RMG [53]. Correlated uncertainties of kinetic and thermodynamic parameters were propagated in a global uncertainty assessment to generate 5000 possible mechanisms for the $CO_2$ methanation on Ni(111). In this global assessment, the uncertainty of kinetic and thermodynamic parameters were quantified by exploiting the Brønsted-Evans-Polanyi (BEP) [54] and linear scaling relations [55,56] from a Pt(111) database with 65 adsorbates [57]. From this exhaustive mechanism exploration

with many hundreds of possible intermediates and pathways, the subset of important chemical steps was identified by sensitivity analyses and comparison with methanation experiments. In this study, only a subset of the sophisticated mechanism from ref. [53], which contains the essential chemistry but can describe $CO_2$ methanation experiments on $Ni/SiO_2$ catalysts [53,58], was included. The subset contains 22 reversible reactions, 5 gas-phase species ($CH_4$, $CO_2$, $H_2O$, $H_2$, CO), and 15 surface species, see Table 1. The reverse reaction rates are calculated via the forward reaction rates and the equilibrium constants (Equation (22)), which are derived from the thermochemistry [53,59]. The mechanism as a Cantera input file including the thermochemistry of the gas and surface species can be found in the Supplementary Information.

**Table 1.** Microkinetic mechanism of the $CO_2$ methanation from ref. [53]. [†] indicate initial sticking coefficients.

| | Reaction | $A$ in ($cm^2 min^{-1} s^{-1}$) or $s$ [†] | $E_a$ in ($kJ mol^{-1}$) |
|---|---|---|---|
| 1 | $CO + {}^* \rightleftharpoons CO^*$ | $0.8$ [†] | 0.0 |
| 2 | $CO_2^* + {}^* \rightleftharpoons CO^* + O^*$ | $4.20 \cdot 10^{19}$ | 74.5 |
| 3 | $CH_4 + {}^* \rightleftharpoons CH_4^*$ | $0.1$ [†] | 0.0 |
| 4 | $CO_2 + {}^* \rightleftharpoons CO_2^*$ | $7.00 \cdot 10^{-3}$ [†] | 0 |
| 5 | $H_2O + {}^* \rightleftharpoons H_2O^*$ | $0.1$ [†] | 0.0 |
| 6 | $H_2 + 2{}^* \rightleftharpoons 2H^*$ | $0.1$ [†] | 17.8 |
| 7 | $OH^* + {}^* \rightleftharpoons H^* + O^*$ | $3.20 \cdot 10^{21}$ | 22.2 |
| 8 | $H_2O^* + {}^* \rightleftharpoons OH^* + H^*$ | $6.40 \cdot 10^{21}$ | 97.1 |
| 9 | $CH_2^* + {}^* \rightleftharpoons CH^* + H^*$ | $6.40 \cdot 10^{21}$ | 0.0 |
| 10 | $CH_3^* + {}^* \rightleftharpoons CH_2^* + H^*$ | $9.60 \cdot 10^{21}$ | 67.0 |
| 11 | $CH_4^* + {}^* \rightleftharpoons CH_3^* + H^*$ | $1.28 \cdot 10^{22}$ | 102.4 |
| 12 | $COOH^* + {}^* \rightleftharpoons OH^* + CO^*$ | $3.20 \cdot 10^{21}$ | 77.5 |
| 13 | $COOH^* + {}^* \rightleftharpoons CO_2^* + H^*$ | $3.20 \cdot 10^{21}$ | 74.0 |
| 14 | $COOH^* + {}^* \rightleftharpoons COH^* + O^*$ | $3.20 \cdot 10^{21}$ | 21.5 |
| 15 | $HCO^* + {}^* \rightleftharpoons H^* + CO^*$ | $3.20 \cdot 10^{21}$ | 0.0 |
| 16 | $HCO^* + {}^* \rightleftharpoons CH^* + O^*$ | $3.20 \cdot 10^{21}$ | 43.7 |
| 17 | $COH^* + {}^* \rightleftharpoons H^* + CO^*$ | $3.20 \cdot 10^{21}$ | 61.5 |
| 18 | $COH^* + O^* \rightleftharpoons OH^* + CO^*$ | $3.20 \cdot 10^{21}$ | 28.8 |
| 19 | $COOH^* + O^* \rightleftharpoons CO_2^* + OH^*$ | $3.20 \cdot 10^{21}$ | 57.0 |
| 20 | $COOH^* + OH^* \rightleftharpoons CO_2^* + H_2O^*$ | $3.20 \cdot 10^{21}$ | 0.0 |

## 3. Results and Discussion

### 3.1. Flow and Species Fields from CFD Simulations

Figure 3 shows streamlines and velocity vector scenes for $\dot{V} = 50$ $ml_N min^{-1}$ for the sphere and ring catalyst pellet, respectively. The flow is from left to right. The streamlines indicate the main flow direction. For the sphere, the flow approaches the particle in the front stagnation point, separates, and unites after passing the cross-shaped holder. These four flow channels can be seen nicely in the back view. The recirculation zones are more visible in the velocity vector scenes in subfigures (B) and (D) in the upper-left region of the reaction chamber. A more complex flow field is obtained for the ring since the inner hole is not perpendicular to the main flow direction. Recirculation zones are also found in the upper-left region of the chamber. For more details on the flow field, as well as results and discussions on the residence time, the reader is referred to the Supporting Information and our previous work [16].

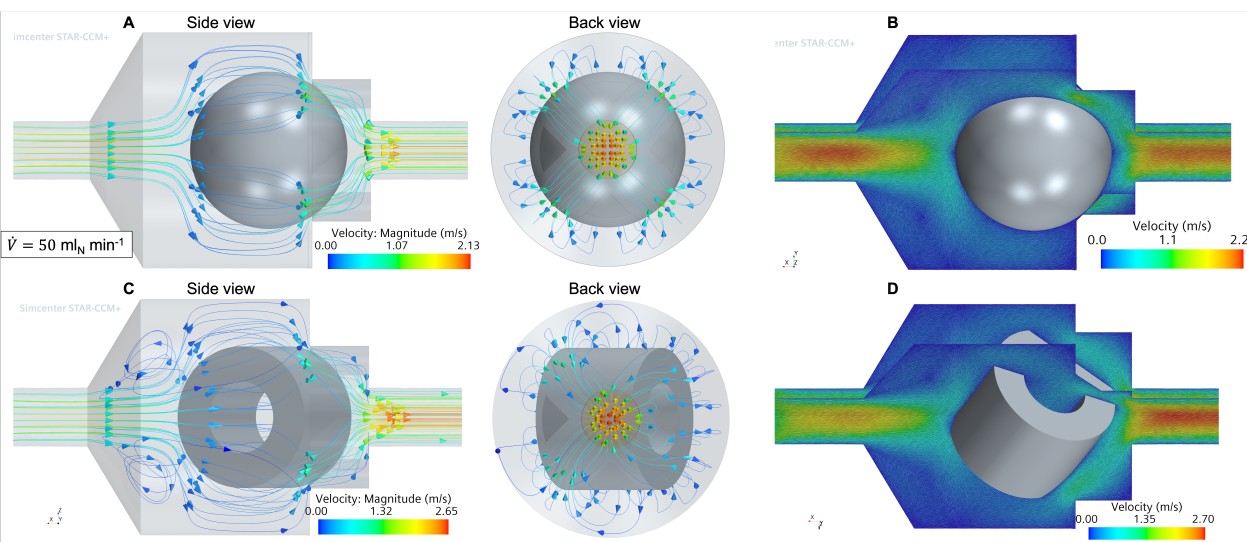

**Figure 3.** (**A**,**C**) streamlines and (**B**,**D**) velocity vector scene for sphere (**top**) and ring (**bottom**) catalyst pellet at $\dot{V} = 50\ \mathrm{ml_N\ min^{-1}}$.

In this study, we focus on the concentration profiles during the catalytic reaction. Gas-phase mole fractions are shown in Figure 4 for $\dot{V} = 200\ \mathrm{ml_N\ min^{-1}}$, exemplary for hydrogen and methane. Fields for other gas-phase species and volumetric flow rates are presented in the SI. Hydrogen and carbon dioxide are consumed over the catalyst surface, whereas methane and water are produced. The third gas-phase product, carbon monoxide, is not shown, but its scalar field is similar to the other two products. For better visualization, isolines of the same mole fractions are shown in black. In general, they are not parallel to the flow direction (from left to right) over the whole length. Curved isolines are found at the entrance region of the chamber, as well as in the rear region of the sphere. However, parallel isolines appear right after the stagnation point until approximately the equator of the sphere. Low reactant and likewise high product mole fraction occur where the sphere approaches the chamber wall. This is due to the fact that the velocity is comparatively low in this area and hence the local residence time is high. For the higher volumetric flow rate, the recirculation zone intensely mixes the fluid from the upper-left region. After the equator of the sphere, the isolines tend to the right-hand side, indicating strong channels. Once again, the touching region between the sphere and the wall results in high conversion of the reactants. For both cases, the fluid is well mixed in the exit tube. Figure 5 shows adsorbed species surface fractions on the sphere for $\dot{V} = 200\ \mathrm{ml_N\ min^{-1}}$. The isolines look the same as latitudes, except for the stagnant spot, where the particle is close to the reactor wall. The surface of the catalyst is mostly covered by CO* with maximum values of approx. 0.23. These figures reveal that the surface coverage is not constant over the entire catalyst particle and is highly dependent on the outer flow conditions.

The situation of gas-phase species distribution for the ring scenario is in general similar to the sphere case. However, the inclined orientation of the ring leads to a more complex situation. For the low flow rate, the isolines are parallel to the flow only in the front region of the ring. Then, the different velocities change the local residence times, which ultimately leads to different local conversion rates. The gas phase is not entirely mixed over the cross-section in the exit tube. For the high flow rate, the situation is even more complex, where in the rear stagnant region the conversion is rather large. This indicates that some part of the fluid is passing rapidly through the chamber, whereas another part is slowly leaving the chamber. The adsorbed species reveal that the fluid flow situation inside the ring is different from the outside. The values of the adsorbed species inside the ring are similar to the values at the rear stagnation region. This indicates lower velocities (and likewise higher local residence times) inside the ring than around it. Once again, the surface is covered by CO*.

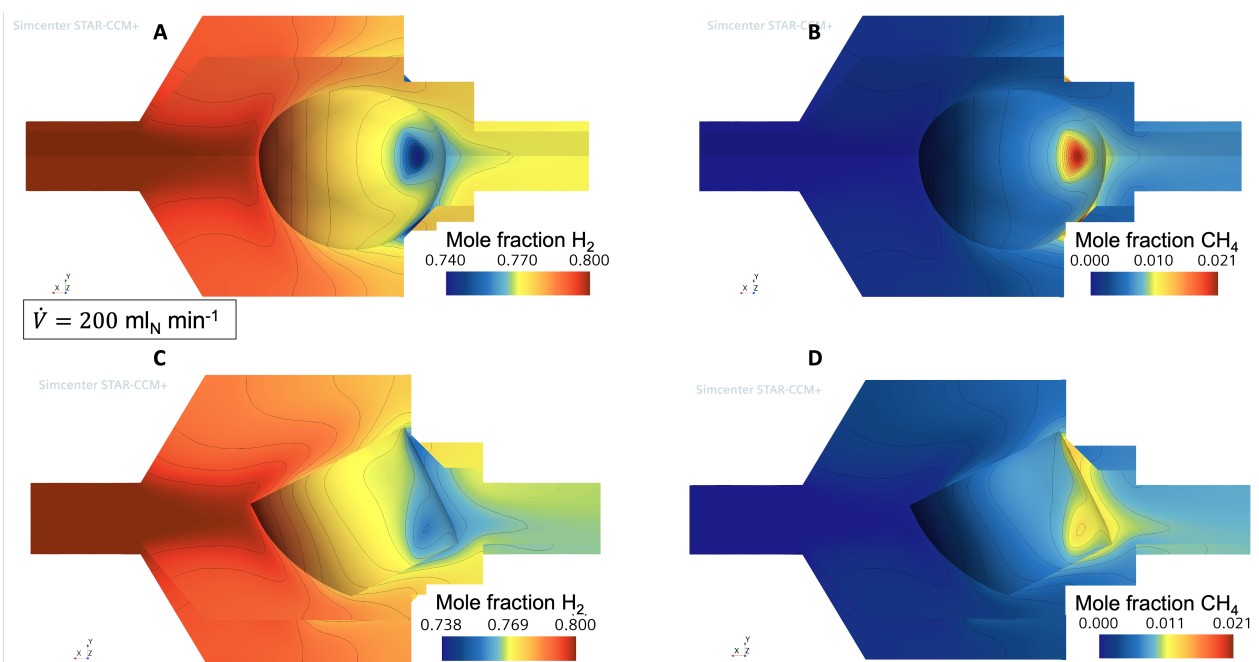

**Figure 4.** Sphere (**top**) and ring (**bottom**) inside Temkin single chamber. Gas-phase mole fractions at $\dot{V} = 200\,\mathrm{ml_N}\,\mathrm{min}^{-1}$: (**A**,**C**) hydrogen, (**B**,**D**) methane.

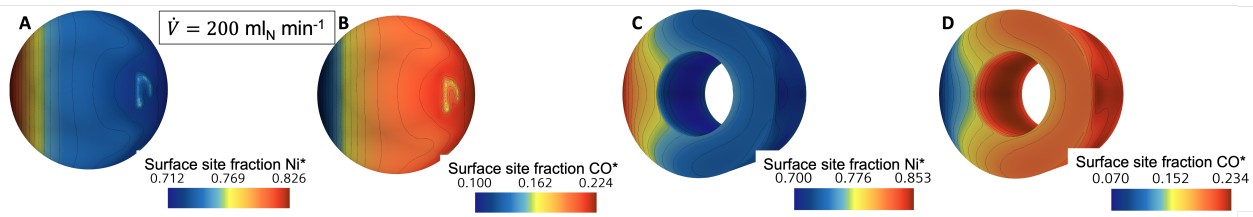

**Figure 5.** Adsorbed species (**A**,**C**) Ni* and (**B**,**D**) CO* on sphere and ring inside Temkin single chamber. $\dot{V} = 200\,\mathrm{ml_N}\,\mathrm{min}^{-1}$.

### 3.2. External Mass Transport Evaluation

In the following, we analyze the dominant regime in terms of mass transport and reaction rate with the local Damköhler number $Da$, which represents the ratio between the reaction rate and the diffusive mass transfer rate. Generally, for $Da \gg 1$, the reaction rate is much faster than mass diffusion. In this regime, mass diffusion is dominant, i.e., the diffusive regime. However, for $Da \ll 1$, the reaction rate (producing or consuming a chemical species) controls the mass transport, i.e., the kinetic regime. Since in this study multiple reaction steps are included in the microkinetic mechanism, Damköhler numbers are formulated for each species with a pseudo first-order kinetic constant $k_i^*$ (m s$^{-1}$):

$$Da_i = \frac{\text{reaction rate}}{\text{diffusive mass transfer rate}} = \frac{k_i^* \cdot L}{D_i^{\mathrm{M}}} = \frac{\dot{s}_i \cdot L}{c_{i,\mathrm{g}} \cdot D_i^{\mathrm{M}}} \tag{32}$$

where $L$ is the characteristic length (m), which is set to be the pellet radius $L = R$, $\dot{s}_i$ is the molar net production rate of species $i$ (mol m$^{-2}$ s$^{-1}$), $c_{i,\mathrm{g}}$ is the bulk gas phase concentration (mol m$^{-3}$), and $D_i^{\mathrm{M}}$ is the effective diffusivity of species $i$ in the mixture M (m$^2$ s$^{-1}$). The Damköhler numbers are evaluated on a line across the pellets, as shown by the plane sections in Figure 4.

In Figure 6, local Damköhler numbers are shown along the particle surface for different volumetric flow rates in the axial direction for the sphere and ring, respectively. Additional figures are presented in the Supporting Information. Noticeably, distinct profiles are seen for the reactants (CH$_4$ and CO$_2$) and the products (H$_2$, H$_2$O, and CO). The local Damköhler

numbers for the reactants are below unity, indicating the kinetic regime for all investigated cases. This is typical for differential reactors since the reactants are not largely consumed. Contrarily, the products Damköhler numbers start in the mass transport regime ($Da > 1$) and decrease with increasing axial distance. Depending on the volumetric flow rate, they either end up in the kinetic regime (for $\dot{V} = 50$ and $100$ ml$_N$ min$^{-1}$) or stay in the mass transport regime (or $\dot{V} = 200$ and $2000$ ml$_N$ min$^{-1}$). The highest value of the local Damköhler number is found in all cases at the front stagnation point of the catalytic pellet, where the fluid boundary layer is the smallest and the reactant concentration is the largest. Since the product Damköhler numbers indicate the diffusive regime, a simplified fluid dynamics model can typically not address the complex species transport.

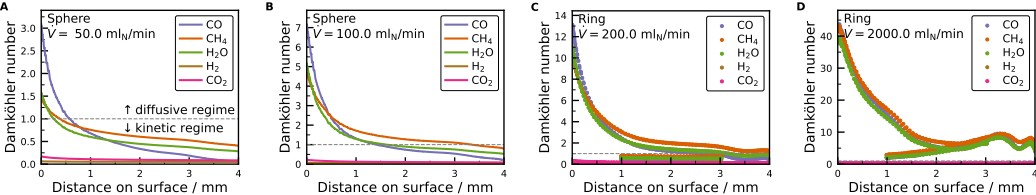

**Figure 6.** Damköhler number for (**A**,**B**) sphere and (**C**,**D**) ring in Temkin single chamber. (**A**) $\dot{V} = 50$ ml$_N$ min$^{-1}$, (**B**) $\dot{V} = 100$ ml$_N$ min$^{-1}$, (**C**) $\dot{V} = 200$ ml$_N$ min$^{-1}$, and (**D**) $\dot{V} = 2000$ ml$_N$ min$^{-1}$.

### 3.3. Comparison between CFD and Simplified Models

At first, the CFD results are compared with the ideal reactor models, see Figure 7 and Supporting Information for additional figures. The axial gas-phase species profiles are similar for the ring and sphere. The profiles of the CFD simulations were obtained on streamlines, as seen in Figure 3. For the PFR model, the calculation domain is separated into an inert, catalytically active, and inert section. It can well predict the exit gas-phase concentration for $\dot{V} = 50$ and $100$ mL$_N$ min$^{-1}$. This is mainly attributed to the presence of the diffusive regime for these low flow rates, cf. Figure 6. Since the catalytic surface is not distributed equally over the chamber length (the sphere starts at 3.6 mm), the rise of the product mole fraction is due to counter-current diffusion. This effect decreases with increasing flow rate. For the two highest flow rates, the PFR model overpredicts the exit product concentration. However, the concentration profiles upstream are not well predicted. This is caused by the reduced available catalytic surface area, but also due to complex fluid flow patterns, counter-current diffusion effects, convective recirculation, as well as short circuiting.

In previous work, we were able to describe the RTD inside a single Temkin reactor chamber with a series of three CSTRs [16]. However in this work, the exit concentrations of the products are overpredicted with 3 CSTRs in series, see star symbols. This impressively highlights that it is sometimes not possible to directly infer the right reactor model from the RTD. Yet, the volume of the eddy in the Temkin segment is quite large, which suggest that back-mixing has a huge influence on the RTD and it does not disagree with our previous study. Rather, it shows the challenges that are entailed when we have completely different flow patterns in a reactor (CSTR and PFR), and the reaction zones are strictly separated. The reaction occurs in the PFR segment in our case, while the CSTR contributes just with physical mixing to the concentration profiles. Theoretically, a transient CRN model should be able to reproduce the correct RTD as well, but this is outside the scope of this work.

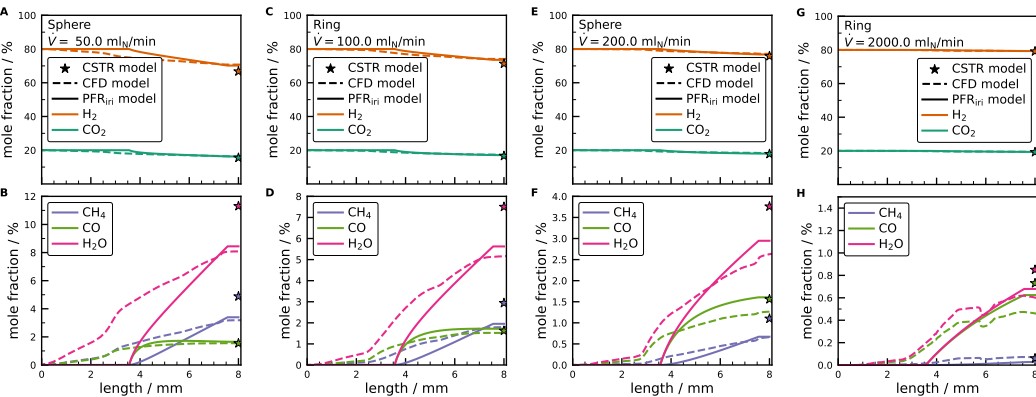

**Figure 7.** Sphere and ring inside the Temkin single chamber. Comparison between 1D PFR model inert–active–inert (PRF$_{\text{iri}}$), tanks-in-series (CSTR) model, and 3D CFD simulations of axial profiles of (**top**) reactants and (**bottom**) products mole fractions. (**A,B**) $\dot{V} = 50$ ml$_N$ min$^{-1}$, (**C,D**) $\dot{V} = 100$ ml$_N$ min$^{-1}$, (**E,F**) $\dot{V} = 200$ ml$_N$ min$^{-1}$, and (**G,H**) $\dot{V} = 2000$ ml$_N$ min$^{-1}$.

Finally, the CFD results are compared with the CRN model, see Figure 8 and Supporting Information for additional figures. Three parameters can be adjusted in the CRN model, i.e., the recycle ratio $R$, the modified available catalyst surface area $F'_{\text{cat/geo}}$, and the length of the non-reactive PFR $L$ at the beginning, see Figure 2B. In order to reduce the complexity of parameter adjustment, we fixed two of the three parameters, i.e., $L$ and $F'_{\text{cat/geo}}$. The position $L$ was set as 2.5 mm, where the typical inclination of the product mole fraction changes from linear to exponential. Although the actual catalyst particles starts at 3.6 mm, an earlier increase in the product mole fractions occurs due to counter-current diffusion processes and back-mixing through recirculation. Since diffusion is neglected in the simplified model, this phenomenon is approximated with back-feeding and enlargement of the catalytic active reaction length. Then, the available catalyst surface area $F'_{\text{cat/geo}}$ is reduced and fixed to the value of 92, although the actual value in the CFD simulations is 100. This value lumps channeling effects, boundary layer effects, and stagnant regions, which cannot be represented explicitly with this simplified 1D model. Finally, the recycle ratio $R$ is fitted in such a way that the slope in the back-feeding region closely matches the CFD results. $R$ values are in the range of 4% to 40%. The higher the volumetric flow rate, the smaller the recycle ratio, see Figure 9. Interestingly, the recycle ratio is very close for the sphere and ring particle shape and both decrease similarly with increasing volumetric flow rate. This behavior originates from the decreasing effect of counter-current diffusion with increasing convective flows, as well as a decreased recirculation zones, see Figure 7 in our previous study [16]. For volumetric flow rates larger than 100 ml$_N$ min$^{-1}$, the recycle ratios change only little. In general, the CRN model gives an excellent agreement with the CFD results. The nearly linear increase in product mole fractions before the catalyst particle, from 0 to 2.5 mm, can be well represented with the back-feeding approach. Then, the reactive wall PFR model can reproduce the detailed CFD results until the end of the particle (length = 7.6 mm). Afterwards, the concentrations do not change significantly. Hence, the non-reactive PFR model is an appropriate approximation. Whereas the CRN agrees well for $\dot{V} = 50$ and $100$ mL$_N$ min$^{-1}$, larger deviations are present for CO and H$_2$O concentrations at $\dot{V} = 200$ ml$_N$ min$^{-1}$. This might originate from the different diffusion coefficients of the product gas-phase species leading to different recycle streams, which is not accounted for in the CRN model. Finally, the fully turbulent case at $\dot{V} = 2000$ ml$_N$ min$^{-1}$ is difficult to predict with the CRN, since complex local mixing patterns occur, which also affect the available surface area. Such three-dimensional flow behavior cannot be represented well with a simple 1D model. Nonetheless, the CRN model is a great improvement in comparison to the ideal CSTR and PFR models. In order to further improve the CRN model, especially for higher flow rates, one might follow the approach from the combustion community and split the calculation domain into a larger set of different zones in at least two-dimensions,

as was demonstrated elsewhere [41]. However, it is worth emphasizing that the typical range of operation of a reactive modular Temkin reactor is below $\dot{V} = 200\ \mathrm{ml_N\ min^{-1}}$; consequently, this CRN approach gives reasonable data in just a few seconds. The CFD model, however, needs around 140 h per single CPU for convergence, which is mainly attributed to the solving of the microkinetic model.

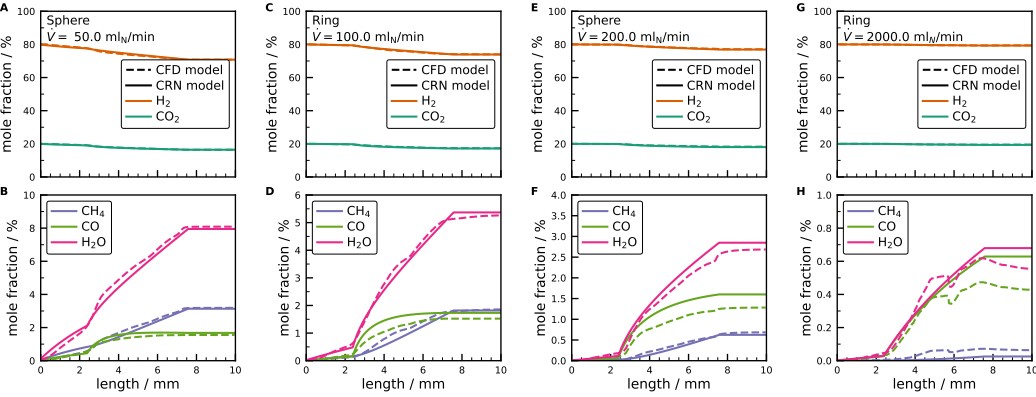

**Figure 8.** Sphere and ring inside Temkin single chamber. Comparison between 1D CRN model and 3D CFD simulations of axial profiles of (**top**) reactants and (**bottom**) products mole fractions. (**A,B**) $\dot{V} = 50\ \mathrm{ml_N\ min^{-1}}$, (**C,D**) $\dot{V} = 100\ \mathrm{ml_N\ min^{-1}}$, (**E,F**) $\dot{V} = 200\ \mathrm{ml_N\ min^{-1}}$, and (**G,H**) $\dot{V} = 2000\ \mathrm{ml_N\ min^{-1}}$.

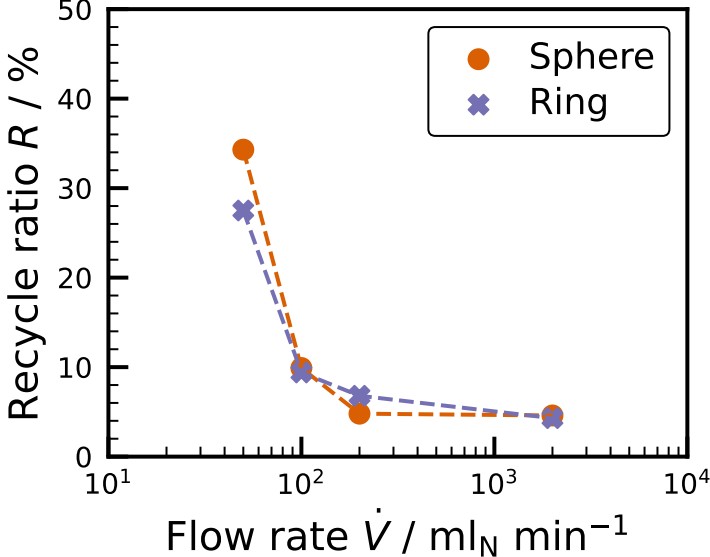

**Figure 9.** Recycle ratio $R$ over volumetric flow rate of the CRN model. $F'_{\mathrm{cat/geo}}$ and $L$ are constant.

## 4. Conclusions

In this work, we theoretically investigated the flow and species fields of a modular Temkin reactor for the catalytic $CO_2$ methanation at isothermal conditions without internal heat and mass transport limitations. Detailed CFD simulations reveal complex interactions between local kinetics and local transport phenomena. Simplified reactor models were evaluated based on the axial species profiles from the CFD simulations. Although in our previous study [16], the RTD of the modular Temkin reactor could be described with 3 CSTRs in series, this model overpredicts the outlet gas concentration. A PFR can also not describe the axial profiles, whereas the largest deviations occur at the beginning of the reaction chamber. Therefore, a CRN model was developed with a combination of non-reactive PFRs, PFRs with a reactive wall, as well as a back-feeding approach. With this CRN model, it is possible to describe the axial concentration profiles inside the reaction

chamber well, especially for the lower flow rates. The most significant model parameter of the CRN is the recycle ratio *R*, which is similar for both particle shapes and decreases with increasing flow rate, while approaching a limit for higher flow rates. The CRN model reaches its limits for the highest flow rate where highly complex flow patterns occur. Since the Temkin reactor is typically operated under low to medium flow rates, this CRN model is appropriate for lab-scale kinetic testing with an emphasis on kinetic mechanism development. Future research should include heat and mass transfer limitations inside the catalyst particles as well as corresponding experimental studies.

**Supplementary Materials:** The following are available online at https://www.mdpi.com/article/10.3390/catal12030349/s1, File S1: Supporting Information PDF. File S2: Python script of the CRN code in Cantera as ZIP file.

**Author Contributions:** Conceptualization, G.D.W., B.K. and C.F.G.; methodology, G.D.W., B.K., and C.F.G.; investigation, G.D.W. and B.K. ; resources, G.D.W., B.K. and C.F.G.; writing—original draft preparation, G.D.W.; writing—review and editing, G.D.W., B.K. and C.F.G.; visualization, G.D.W.; project administration, G.D.W. and B.K.; funding acquisition, B.K. and C.F.G. All authors have read and agreed to the published version of the manuscript.

**Funding:** BK and CFG gratefully acknowledge support by the U.S. Department of Energy, Office of Science, Basic Energy Sciences, under Award #0000232253, as part of the Computational Chemical Sciences Program.

**Acknowledgments:** We thank Anton Nagy from ILS (Berlin, Germany) for sharing the CAD files of their modular Temkin reactor. We acknowledge support by Open Access Publishing Fund of Clausthal University of Technology.

**Conflicts of Interest:** The authors declare no conflict of interest.

## Abbreviations

The following abbreviations are used in this manuscript:

| | |
|---|---|
| CFD | Computational Fluid Dynamics |
| CRN | Chemical reactor network |
| CSTR | Continuously stirred tank reactor |
| iri | inert–active–inert |
| PFR | Plug flow reactor |
| RTD | Residence time distribution |

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
