# Peer review of "Non-Idealities in Lab-Scale Kinetic Testing: A Theoretical Study of a Modular Temkin Reactor"

_catalysts, doi:10.3390/catal12030349_

Round 1
Reviewer 1 Report
Attached

Author Response
Reviewer 1:
Remark 1.1: This study compares one-dimensional models with CFD simulation of a Temkin reactor. Although the authors bring out the importance of back mixing in this kind of reactor, I don’t
think the paper is scientifically very sound. The major problem I see with the whole study is that the recycle ratio is a parameter that is adjusted to match the results with that of the CFD simulation. I would have greatly appreciated this work if the authors had developed a methodology for predicting the recycle ratio so that the reactor networks models could be used with confidence. For every single flow rate, the recycle ratio changes. It’s not practical to set these recycle ratios by performing CFD simulations. Nevertheless, for the given problem definition, the authors demonstrate the superiority of the reactor network model with recycling rectors.
Answer 1.1: Admittedly, the fitting of the recycle ratio to the CFD simulation is a postdictive approach and a a priori determination of the recycle ratio should be aimed for. However, this is a very complicated problem. The recycle ratio of the developed reactor network model “lumps” the non-idealities of flow in this Temkin reactor, i.e., deviation from plug-flow due to channeling, recirculation, flow detachment, etc. It was shown with the detailed CFD simulations, that the local velocity fields in the reactor chamber changes with flow rate. Consequently, it is not surprising that the recycle ratio changes, too. The dimensions and shape of the eddy close to the inlet depends on the flow conditions and therefore affects the degree of recirculation. We agree that is not practical to derive the recycle ratios of every flow condition and many other particle shapes only with detailed CFD simulations. Since we do not have experimental concentration maps from inside the reaction chamber, this study is only based on simulations. We wish to highlight that this is the first study, to actually unveil that the complex flow pattern can be described by such chemical network approach. The combination of CFD with the CRN method can now be systematically used to investigate relations between the recycle ratio, flow rate, shape and catalyst geometry to derive a predictive relation for the recycle ratio.
Action 1.1: None.
Remark 1.2: The authors say that the steady laminar formulation is presented for brevity sake. The readers then get carried away with the impression that the simulations address only laminar flow cases. However, the authors also did one set of turbulent case studies. Therefore, this sentence must be rephrased accordingly.
Answer 1.2: We agree that presenting the laminar case might be misleading. We therefore present now the turbulent RANS formulation in section 2.1.1. Governing equations.
Action 1.2: Please see reworked section 2.1.1. on page 3 with turbulent formulation in Eq. (1-13): “The turbulent Reynolds Averaged Navier Stokes (RANS) equations read: …”
Remark 1.3: It is not clear why a CVode solver is required to calculate reaction rates as they are all algebraic equations.
Action 1.3: We added the following sentence to section 2.1.3 on page 6: “In Simcenter STAR-CCM+, the reacting species transport equations are solved using the CVODE solver with the operator splitting algorithm to find an average reaction rate to remove stiffness.”
Remark 1.4: How is the source terms calculated for the estimation of Damkholer number? Source term changes over the surface of the catalyst surface? What sort of averaging are you performing?
Answer 1.4: We evaluated the Damköhler number on lines on the pellets, as seen in Fig. 4. Since the stagnation zones cover only a small area on the surface, we did not include this area in the Damköhler number evaluation.
Action 1.4: We added to section 3.2: “The Damköhler numbers are evaluated on a line across the pellets, as shown by the plane sections in Fig. 4”.
Remark 1.5: What is the Fcat/geo factor used for plug flow reactor mode? Is it constant for all flow rates?
We kept the available reactive surface area constant in all models. The Fcat/geo was 100 as in the CFD simulations.
Action 1.5: None.
Remark 1.6: How is the length of the non-reacting PFR entering into the CRN model?
Answer 1.6: The length of the non-reacting PFR is set constant to 2.5 mm. In this section, the back-feeding is performed. If for example the length of the non-reacting PFR is set to zero, no back-feeding occurs, and the system behaves like a reactive PFR.
Action 1.6: None.
Reviewer 2 Report
This is an excellent manuscript and should be published. The work addresses the residence time distribution and more generally non-ideal behaviour of a Temkin reactor, a laboratory reactor used for catalyst testing and kinetic measurements using original industrial catalyst particle dimensions. Because of its laboratory scale and its geometric features (a string of catalyst pellets in sequential but separated chambers of), it is assumed that Temkin reactors behave like a series of CSTRs, perfectly mixed tank reactors. However, this study demonstrates a quite different more complex behaviour of a Temkin reactor. The authors show this using rigorous CFD simulations combined with a quite complex microkinetic model of a very detailed reaction mechanism, a very challenging and novel approach. Furthermore, besides its suitability for such studies due to its high reaction rates, the chosen reaction system of CO2 methanation is of high industrial relevance. Very importantly, the authors are not restricted in demonstrating the non-idealities of a Temkin reactor but develop a suitable reactor network model to describe its behaviour.
The CFD studies implementing a complex and detailed microkinetic model are novel and meticulously executed. The manuscript is well written with numerous novel results. The importance of the work is vast as it proves the non-ideal behaviour of the considered reactor and points to the right direction for future kinetic and industrial catalyst testing.
I would recommend the publication of the manuscript as it is.
Author Response
We thank this reviewer for his/her time and the positive assessment.